# Multiclass versus Binary Differentially Private PAC Learning

**Mark Bun**
Department of Computer Science
Boston University
Boston, MA 02215
mbun@bu.edu

**Marco Gaboardi**
Department of Computer Science
Boston University
Boston, MA 02215
gaboardi@bu.edu

**Satchit Sivakumar**
Department of Computer Science
Boston University
Boston, MA 02215
satchit@bu.edu

## Abstract

We show a generic reduction from multiclass differentially private PAC learning to binary private PAC learning. We apply this transformation to a recently proposed binary private PAC learner to obtain a private multiclass learner with sample complexity that has a polynomial dependence on the multiclass Littlestone dimension and a poly-logarithmic dependence on the number of classes. This yields a doubly exponential improvement in the dependence on both parameters over learners from previous work. Our proof extends the notion of $\Psi$-dimension defined in work of Ben-David et al. [5] to the online setting and explores its general properties.

## 1 Introduction

[1] Machine learning and data analytics are increasingly deployed on sensitive information about individuals. Differential privacy [10] gives a mathematically rigorous way to enable such analyses while guaranteeing the privacy of individual information. The model of *differentially private PAC learning* [17] captures binary classification for sensitive data, providing a simple and broadly applicable abstraction for many machine learning procedures. Private PAC learning is now reasonably well-understood, with a host of general algorithmic techniques, lower bounds, and results for specific fundamental concept classes [8, 12, 3, 4, 1, 15, 2, 16].

Beyond binary classification, many problems in machine learning are better modeled as *multiclass learning* problems. Here, given a training set of examples from domain $\mathcal{X}$ with labels from $[k] = \{0, 1, \ldots, k\}$, the goal is to learn a function $h : \mathcal{X} \to [k]$ that approximately labels the data and generalizes to the underlying population from which it was drawn. Much less is presently known about differentially private multiclass learnability than is known about private binary classification, though it appears that many specific tools and techniques can be adapted one at a time. In this work, we ask: *Can we generically relate multiclass to binary learning so as to automatically transfer results from the binary setting to the multiclass setting?*

To illustrate, there is a simple reduction from a given multiclass learning problem to a sequence of binary classification problems. (This reduction was described by Ben-David et al. [5] for non-private

---

[1]The full version of this paper with all of the details can be found at https://arxiv.org/abs/2107.10870

35th Conference on Neural Information Processing Systems (NeurIPS 2021).

learning, but works just as well in the private setting.) Intuitively, one can learn a multi-valued label one bit at a time. That is, to learn an unknown function $f : \mathcal{X} \to [k]$, it suffices to learn the $\lceil \log_2(k+1) \rceil$ binary functions $f_i : \mathcal{X} \to [k]$, where each $f_i$ is the $i^{th}$ bit of $f$.

**Theorem 1.1** (Informal). *Let $H$ be a concept class consisting of $[k]$-valued functions. If all of the binary classes $H_i = \{f_i : f \in H\}$ are privately learnable, then $H$ is privately learnable.*

Beyond its obvious use for enabling the use of tools for binary private PAC learning on the classes $H_i$, we show that Theorem 1.1 has strong implications for relating the private learnability of $H$ to the combinatorial properties of $H$ itself. Our main application of this reductive perspective is an improved sample complexity upper bound for private multiclass learning in terms of online learnability.

## 1.1 Online vs. Private Learnability

A recent line of work has revealed an intimate connection between differentially private learnability and learnability in Littlestone's mistake-bound model of online learning [18]. For binary classes, the latter is tightly captured by a combinatorial parameter called the Littlestone dimension; a class $H$ is online learnable with mistake bound at most $d$ if and only if its Littlestone dimension is at most $d$. The Littlestone dimension also qualitatively characterizes private learnability. If a class $H$ has Littlestone dimension $d$, then every private PAC learner for $H$ requires at least $\Omega(\log^* d)$ samples [1]. Meanwhile, Bun et al. [7] showed that $H$ is privately learnable using $2^{2^{O(d)}}$ samples, and Ghazi et al. [13] gave an improved algorithm using $\tilde{O}(d^6)$ samples. (Moreover, while quantitatively far apart, both the upper and lower bound are tight up to polynomial factors as functions of the Littlestone dimension alone [15].)

Jung et al. [14] recently extended this connection from binary to multiclass learnability. They gave upper and lower bounds on the sample complexity of private multiclass learnability in terms of the *multiclass Littlestone dimension* [9]. Specifically, they showed that if a multi-valued class $H$ has multiclass Littlestone dimension $d$, then it is privately learnable using $2^{k^{O(d)}}$ samples and that every private learner requires $\Omega(\log^* d)$ samples.

Jung et al.'s upper bound [14] directly extended the definitions and arguments from Bun et al.'s [7] earlier $2^{2^{O(d)}}$-sample algorithm for the binary case. While plausible, it is currently unknown and far from obvious whether similar adaptations can be made to the improved binary algorithm of Ghazi et al. [13]. Instead of attacking this problem directly, we show that Theorem 1.1, together with additional insights relating multiclass and binary Littlestone dimensions, allows us to *generically* translate sample complexity upper bounds for private learning in terms of binary Littlestone dimension into upper bounds in terms of multiclass Littlestone dimension. Instantiating this general translation using the algorithm of Ghazi et al. gives the following improved sample complexity upper bound.

**Theorem 1.2** (Informal). *Let $H$ be a concept class consisting of $[k]$-valued functions and let $d$ be the multiclass Littlestone dimension of $H$. Then $H$ is privately learnable using $\tilde{O}(d^6 \log^8(k+1))$ samples.*

In addition to being conceptually simple and modular, our reduction from multiclass to binary learning means that potential future improvements for binary learning will also automatically give improvements for multiclass learning. For example, if one were able to prove that all binary classes of Littlestone dimension $d$ are privately learnable with $O(d)$ samples, this would imply that every $[k]$-valued class of multiclass Litttlestone dimension $d$ is privately learnable with $\tilde{O}(d \log^3(k+1))$ samples.[2]

## 1.2 Techniques

Theorem 1.1 shows that a multi-valued class $H$ is privately learnable if all of the binary classes $H_i$ are privately learnable, which in turn holds as long as we can control their (binary) Littlestone dimensions. So the last remaining step in order to conclude Theorem 1.2 is to show that if $H$ has bounded multiclass Littlestone dimension, then all of the classes $H_i$ have bounded binary Littlestone dimension. At first glance, this may seem to follow immediately from the fact that (multiclass)

---

[2]The nearly cubic dependence on $\log(k+1)$ follows from the fact that the accuracy of private learners can be boosted with a sample complexity blowup that is nearly inverse linear in the target accuracy [11, 6].

Littlestone dimension characterizes (multiclass) online learnability – a mistake bounded learner for a multiclass problem is, in particular, able to learn each individual output bit of the function being learned. The problem with this intuition is that the multiclass learner is given more feedback from each example, namely the entire multi-valued class label, than a binary learner for each $H_i$ that is only given a single bit. Nevertheless, we are still able to use combinatorial methods to show that multiclass online learnability of a class $H$ implies online learnability of all of the binary classes $H_i$.

**Theorem 1.3.** *Let $H$ be a $[k]$-valued concept class with multiclass Littlestone dimension $d$. Then every binary class $H_i$ has Littlestone dimension at most $6d \ln(k + 1)$.*

Moreover, this result is nearly tight. In the full version of the paper, we show that for every $k, d \geq 1$ there is a $[k]$-valued class with multiclass Littlestone dimension $d$ such that at least one of the classes $H_i$ has Littlestone dimension at least $\Omega(d \log(k + 1))$.

In addition, it turns out that online learnability of the binary classes $H_i$ implies multiclass online learnability of $H$. In the full version of the paper, we show that if the multiclass Littlestone dimension of $H$ is $d$, then there is at least one binary class $H_i$ with Littlestone dimension larger than $\frac{d}{\log(k+1)}$. This result is also tight.

Theorem 1.3 is the main technical contribution of this work. The proof adapts techniques introduced by Ben-David et al. [5] for characterizing the sample complexity of (non-private) multiclass PAC learnability. Specifically, Ben-David et al. introduced a family of combinatorial dimensions, parameterized by collections of maps $\Psi$ and called $\Psi$-dimensions, associated to classes of multi-valued functions. One choice of $\Psi$ corresponds to the "Natarajan dimension" [19], which was previously known to give a lower bound on the sample complexity of multiclass learnability. Another choice corresponds to the "graph dimension" [19] which was known to give an upper bound. Ben-David et al. gave conditions under which $\Psi$-dimensions for different choices of $\Psi$ could be related to each other, concluding that the Natarajan and graph dimensions are always within an $O(\log(k + 1))$ factor, and thus characterizing the sample complexity of multiclass learnability up to such a factor.

Our proof of Theorem 1.3 proceeds by extending the definition of $\Psi$-dimension to online learning. We show that one choice of $\Psi$ corresponds to the multiclass Littlestone dimension, while a different choice corresponds to the maximum Littlestone dimension of any binary class $H_i$. We relate the two quantities up to a logarithmic factor using a new variant of the Sauer-Shelah-Perles Lemma for the "0-cover numbers" of a class of multi-valued functions. While we were originally motivated by privacy, we believe that Theorem 1.3 and the toolkit we develop for understanding online $\Psi$-dimensions may be of broader interest in the study of (multiclass) online learnability.

Finally, we remark that Theorem 1.3 implies a qualitative converse to Lemma 1.1. If a multi-valued class $H$ is privately learnable, then the lower bound of [14] implies that $H$ has finite multiclass Littlestone dimension. Theorem 1.3 then shows that all of the classes $H_i$ have finite binary Littlestone dimension, which implies via [7, 13] that they are also privately learnable.

**Societal Impact**   Our work is motivated by privacy-respecting data analysis. Our focus is on theoretical questions aimed at uncovering general principles about when private learning is feasible. As such, it does not negatively impact the way privacy-respecting techniques are used, but rather it clarifies their potential.

## 2   Background

**Differential privacy.**   Differential privacy is a property of a randomized algorithm guaranteeing that the distributions obtained by running the algorithm on two datasets differing for one individual's data are indistinguishable up to a multiplicative factor $e^\epsilon$ and an additive factor $\delta$. Formally, it is defined as follows:

**Definition 2.1** (Differential privacy, [10])**.** *Let $n \in \mathbb{N}$. A randomized algorithm $M : \mathcal{X}^n \to \mathcal{Y}$ is $(\epsilon, \delta)$-differentially private if for all subsets $S \subseteq \mathcal{Y}$ of the output space, and for all datasets $X$ and $X'$ containing $n$ elements of the universe $\mathcal{X}$ and differing in at most one element (we call these neighbouring datasets), we have that*

$$\Pr(M(X) \in S) \leq e^\epsilon \Pr(M(X') \in S) + \delta$$

One useful property of differential privacy that we will use is that any output of a differentially private algorithm is closed under 'post-processing', that is, its cannot be made less private by applying any data-independent transformations.

**Lemma 2.2** (Post-processing of differential privacy, [10]). *If $M : \mathcal{X}^n \to \mathcal{Y}$ is $(\epsilon, \delta)$-differentially private, and $\mathcal{B} : \mathcal{Y} \to \mathcal{Z}$ is any randomized function, then the algorithm $\mathcal{B} \circ M$ is $(\epsilon, \delta)$-differentially private.*

**PAC learning.** PAC learning [21] aims at capturing natural conditions under which an algorithm can approximately learn an hypothesis class.

**Definition 2.3** (Hypothesis class). *A hypothesis class $H$ with input space $\mathcal{X}$ and output space $\mathcal{Y}$ (also called the label space) is a set of functions $f$ mapping $\mathcal{X}$ to $\mathcal{Y}$.*

Where it is clear, we will not explicitly name the input and output spaces. We can now formally define PAC learning.

**Definition 2.4** (PAC learning, [21]). *A learning problem is defined by a hypothesis class $H$. For any distribution $P$ over the input space $\mathcal{X}$, consider $n$ independent draws $x_1, x_2, \cdots x_n$ from distribution $P$. A labeled sample of size $n$ is the set $\{(x_1, f(x_1)), (x_2, f(x_2)), \cdots, (x_n, f(x_n))\}$ where $f \in H$. We say an algorithm $A$ taking a labeled sample $X$ of size $n$ is an $(\alpha, \beta)$-accurate PAC learner for the hypothesis class $H$ if for all functions $f \in H$ and for all distributions $P$ over the input space, $A$ on being given a labeled sample of size $n$ drawn from $P$ and labeled by $f$, outputs a hypothesis $h \in H$ such that with probability greater than or equal to $1 - \beta$ over the randomness of the sample and the algorithm,*

$$\Pr[h(x) \neq f(x)] \leq \alpha.$$

The definition above defines PAC learning in the *realizable* setting, where all the functions $f$ labeling the data are in $H$. Two well studied settings for PAC learning are the *binary learning* case, where $\mathcal{Y} = \{0, 1\}$ and the *multiclass learning* case, where $\mathcal{Y} = [k] = \{0, 1, \cdots, k\}$ for natural numbers $k > 2$. The natural notion of complexity for PAC learning is *sample complexity*.

**Definition 2.5** (Sample complexity). *The sample complexity $S_{H,\alpha,\beta}(A)$ of algorithm $A$ with respect to hypothesis class $H$ is the minimum size of the sample that the algorithm requires in order to be an $(\alpha, \beta)$-accurate PAC learner for $H$. The PAC complexity of the hypothesis class $H$ is*

$$\inf_A S_{H,\alpha,\beta}(A).$$

In this work, we will be interested in *generic* learners, that work for every hypothesis class.

**Definition 2.6** (Generic learners). *We say that an algorithm $A$ that additionally takes the hypothesis class as an input, is a **generic** $(\alpha, \beta)$-accurate private PAC learner with sample complexity function $SC(H, \alpha, \beta)$, if for every hypothesis class $H$, it is an $(\alpha, \beta)$-accurate private PAC learner for $H$ with sample complexity $SC(H, \alpha, \beta)$.*

**Differentially private PAC learning.** We can now define differentially private PAC learning, by putting together the constraints imposed by differential privacy and PAC learning, respectively.

**Definition 2.7** (Differentially private PAC learning [17]). *An algorithm $A$ is an $(\epsilon, \delta)$-differentially private and $(\alpha, \beta)$-accurate private PAC learner for the hypothesis class $H$ with sample complexity $n$ if and only if:*

1. *$A$ is an $(\alpha, \beta)$-accurate PAC learner for the hypothesis class $H$ with sample complexity $n$.*

2. *$A$ is $(\epsilon, \delta)$-differentially private.*

In this work, we study the complexity of private PAC learning. Our work focuses on the **multiclass realizable** setting.

**Multiclass Littlestone dimension.** We recall here the definition of multiclass Littlestone dimension [9], which we will use extensively in this work. Unless stated otherwise, we will use the convention that the root of a tree is at depth 0. As a first step, we define a class of labeled binary trees, representing possible input-output label sequences over an input space $\mathcal{X}$ and the label space $[k]$.

**Definition 2.8** (Complete io-labeled binary tree). *A complete io-labeled binary tree of depth b with input set $\mathcal{X}$ and output set $[k]$ consists of a complete binary tree of depth b with the following properties:*

1. *Every node of the tree other than the leaves is labeled by an example $x \in \mathcal{X}$.*

2. *The 2 edges going from any parent node to its two children are labeled by two different labels in $[k]$.*

3. *The leaf nodes of the tree are unlabeled.*

We are interested in whether the input-ouput labelings defined by the complete io-labeled tree can be achieved by some function in the hypothesis class; to this end, we define realizability for root-to-leaf paths.

**Definition 2.9.** *Given a complete io-labeled binary tree of depth b, consider a root-to-leaf path described as an ordered sequence $S = \{(x_i, y_i) \,|\, i \in [b]\}$, where $x_i$ is a node label and $y_i$ is the label of the edge between $x_i$ and $x_{i+1}$, and where $x_0$ is the root. We say that the root-to-leaf path is realized by a function $f \in H$ if for every $(x_i, y_i)$ in S, we have $x_i \in \mathcal{X}$ and $y_i = f(x_i)$.*

Using this definition we can now define what it means for a hypothesis class of functions to shatter a complete io-labeled binary tree, which helps to capture how expressive the hypothesis class is.

**Definition 2.10** (Shattering). *We say that a complete io-labeled binary tree of depth b with label set $[k]$ is* shattered *by a hypothesis class H if for all $2^b$ root-to-leaf sequences S of the tree, there exists a function $f \in H$ that realizes S.*

Using this definition of shattering we can finally define the multiclass Littlestone dimension.

**Definition 2.11** (Multiclass Littlestone dimension, [9]). *The **multiclass Littlestone dimension** of a hypothesis class H, denoted $MLS(H)$, is defined to be the maximum b such that there exists a complete io-labeled binary tree of depth b that is shattered by H. If no maximum exists, then we say that the multiclass Littlestone dimension of H is $\infty$.*

## 3 Main results

### 3.1 Reduction from multiclass private PAC learning to binary private PAC learning

Our first main result is a reduction from multiclass private PAC learning to binary private PAC learning. Informally, the idea is that that every function $f$ mapping examples to labels in $[k]$ can be thought of as a vector of binary functions $(f_1, \cdots, f_{\log(k+1)})$. Here, each binary function predicts a bit of the binary representation of the label predicted by $f$. Then, we can learn these binary functions by splitting the dataset into $\log(k + 1)$ parts, and using each part to learn a different $f_i$. We can learn the binary functions using an $(\epsilon, \delta)$-DP binary PAC learner. Then, we can combine the binary hypotheses obtained to get a hypothesis for the multiclass setting, by applying a binary to decimal transformation. This process, described in Figure 1, preserves privacy since changing a single element of the input dataset changes only one of the partitions, and we apply an $(\epsilon, \delta)$-DP learning algorithm to each partition. The binary to decimal transformation can be seen as post-processing.

Next, we formalize this idea. Given a hypothesis class $H$ with label set $[k]$, construct the following $\log(k + 1)$ hypothesis classes $H|_1, \cdots, H|_{\log(k+1)}$. For every function $f \in H$, let $f_i : \mathcal{X} \to \{0, 1\}$ be the function defined such that $f_i(x)$ is the $i^{th}$ bit of the binary expansion of $f(x)$. Let the hypothesis class $H|_i$ be defined as $\{f_i : f \in H\}$. We will call these the **binary restrictions** of $H$.

**Theorem 3.1.** *Let H be a hypothesis class with label set $[k]$ and let $H|_1, \cdots, H|_{\log(k+1)}$ be its binary restrictions. Assume we have $(\epsilon, \delta)$-differentially private, $(\alpha, \beta)$-accurate PAC learners $B^1, \cdots, B^{\log(k+1)}$ for $H|_1, \cdots, H|_{\log(k+1)}$ with sample complexities upper bounded by $SC^1_{\alpha,\beta}, SC^2_{\alpha,\beta}, \cdots, SC^{\log(k+1)}_{\alpha,\beta}$. Then, there exists an $(\epsilon, \delta)$-differentially private, $(\alpha, \beta)$-accurate PAC learner A for the hypothesis class H that has sample complexity upper bounded by $\sum_{i=1}^{\log(k+1)} SC^i_{\alpha/\log(k+1),\beta/\log(k+1)}$.*

The proof of this lemma can be found in the full version of the paper. Next, we recall that the sample complexity of privately learning binary hypothesis classes is characterized by the Littlestone

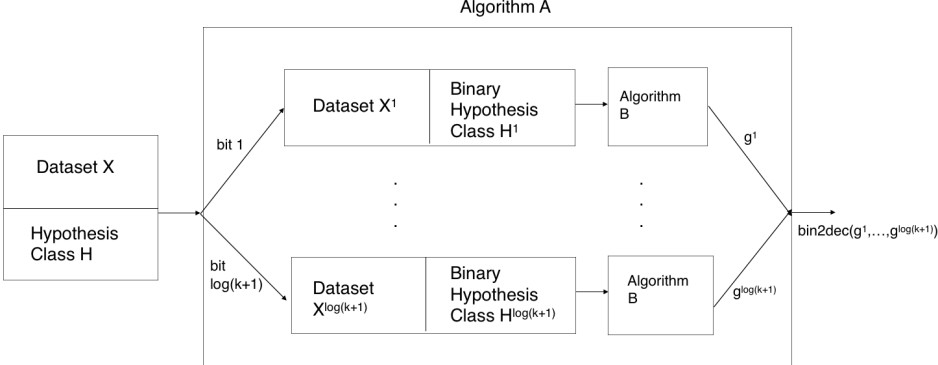

Figure 1: Algorithm $A$ is the $(\epsilon, \delta)$-DP PAC learner for hypothesis classes with label set [k]. The algorithm $B$ used as a subroutine is an $(\epsilon, \delta)$-DP PAC learner for binary hypothesis classes. bin2dec represents a binary to decimal conversion.

dimension of the hypothesis class [1] [7]. That is, there exists an $(\alpha, \beta)$-accurate $(\epsilon, \delta)$-DP PAC learning algorithm for any binary hypothesis class $H$ with sample complexity upper and lower bounded by a function only depending on $\alpha, \beta, \epsilon, \delta$ and $d$ where $d$ is the Littlestone dimension of $H$. Using this characterization, we obtain the following corollary to Theorem 3.1.

**Corollary 3.2.** *Let $H$ be a hypothesis class with label set $[k]$ and let $H|_1, \cdots, H|_{\log(k+1)}$ be its binary restrictions. Let the Littlestone dimensions of $H|_1, \cdots, H|_{\log(k+1)}$ be $d_1, \cdots, d_{\log(k+1)}$. Assume we have a generic $(\epsilon, \delta)$-differentially private, $(\alpha, \beta)$-accurate PAC learner $B$ for binary hypothesis classes $G$ that has sample complexity upper bounded by a function $SC_{\epsilon, \delta}(d', \alpha, \beta)$ where $d'$ is the Littlestone dimension of $G$. Then, there exists an $(\epsilon, \delta)$-differentially private, $(\alpha, \beta)$-accurate PAC learner $A$ for $H$ that has sample complexity upper bounded by $\sum_{i=1}^{\log(k+1)} SC_{\epsilon, \delta}(d_i, \alpha/\log(k+1), \beta/\log(k+1))$.*

Corollary 3.2 shows that the sample complexity of privately PAC learning a hypothesis class in the multiclass setting can be upper bounded by a function depending on the Littlestone dimensions of its binary restrictions. However, as described earlier, Jung et al. [14] showed that the sample complexity of private multiclass PAC learning could be characterized by the multiclass Littlestone dimension. Hence, an immediate question is what the relationship between the multiclass Littlestone dimension of a class and the Littlestone dimensions of its binary restrictions is.

## 3.2 Connection between multiclass and binary Littlestone dimension

We show that the multiclass Littlestone dimension $MLS(H)$ of a hypothesis class is intimately connected to the maximum Littlestone dimension over its binary restrictions.

**Theorem 3.3.** *Let $H$ by a hypothesis class with input set $\mathcal{X}$ and output set $[k]$. Let the multiclass Littlestone dimension of $H$ be $d$. Let $H|_1, H|_2, \cdots, H|_{\log(k+1)}$ be the binary restrictions of $H$. Let the Littlestone dimensions of $H|_1, H|_2, \cdots, H|_{\log(k+1)}$ be $d_1, \cdots, d_{\log(k+1)}$. Then,*

$$\max_{i=1,\cdots,\log(k+1)} d_i \leq 6d\ln(k+1).$$

A similar-looking theorem relating the Natarajan dimension of a hypothesis class with the maximum VC dimension over its binary restrictions was proved in Ben-David et al. [5] using the notion of $\Psi$-dimension. Our proof of Theorem 3.3 is inspired by this strategy. It will proceed by defining and analyzing a notion of dimension that we call $\Psi$-Littlestone dimension. It will also use the 0-cover function of a hypothesis class defined in Rakhlin et al. [20]. The details of the proof are described in Section 5. Finally, combining Theorem 3.3 and Corollary 3.2, we can obtain the following corollary to Theorem 3.1.

**Corollary 3.4.** *Assume we have a generic $(\epsilon, \delta)$-differentially private, $(\alpha, \beta)$-accurate PAC learner $B$ for binary hypothesis classes $G$ that has sample complexity upper bounded by a function*

$SC_{\epsilon,\delta}(d', \alpha, \beta)$ where $d'$ is the Littlestone dimension of $G$. Then, there exists a generic $(\epsilon, \delta)$-differentially private, $(\alpha, \beta)$-accurate PAC learner $A$ for multi-valued hypothesis classes $H$ (label set $[k]$) that has sample complexity upper bounded by $\log(k + 1)SC_{\epsilon,\delta}(6d \log(k + 1), \alpha/\log(k + 1), \beta/\log(k + 1))$ where $d$ is the multiclass Littlestone dimension of $H$.

We now consider an application of this result. The best known sample complexity bound for $(\epsilon, \delta)$-DP binary PAC learning is achieved by a learner described in Ghazi et al. [13]. We state a slightly looser version of their result here.

**Theorem 3.5** (Theorem 6.4 [13])**.** *Let $G$ be any binary hypothesis class with Littlestone dimension $d_L$. Then, for any $\epsilon, \delta, \alpha, \beta \in [0, 1]$, for some*

$$n = O\left(\frac{d_L^6 \log^2(\frac{d_L}{\alpha\beta\epsilon\delta})}{\epsilon\alpha^2}\right),$$

*there is an $(\epsilon, \delta)$-differentially private, $(\alpha, \beta)$-accurate PAC learning algorithm $B$ for $G$ with sample complexity upper bounded by $n$.*

The above learner can be boosted via a technique of Bun et al. [6] to improve the dependence on $\alpha$ to nearly inverse linear. The details of this boosting are discussed in the full version of the paper. Applying the reduction described in Theorem 3.1, with this boosted learner as a subroutine, and using Corollary 3.4, we get the following theorem.

**Theorem 3.6.** *Let $H$ be a concept class over $\mathcal{X}$ with label set $[k]$ and multiclass Littlestone dimension $d$. Then, for any $\epsilon, \delta, \alpha, \beta \in [0, 1]$, for some*

$$n = O\left(\frac{d^6(\log(k + 1))^8 \log^2(\frac{d \log^3 k}{\epsilon\delta\alpha\beta})}{\epsilon\alpha^2}\right)$$

*there is an $(\epsilon, \delta)$-differentially private, $(\alpha, \beta)$-accurate PAC learning algorithm $A$ for $H$ with sample complexity upper bounded by $n$.*

## 4 $\Psi$-Littlestone dimension

### 4.1 Definition

In this section, we define an online analogue to the $\Psi$-dimension [5] that will help us prove Theorem 3.3. The main intuition is that similar to in the definition of $\Psi$-dimension, we can use what we term *collapsing maps* to reason about the multiclass setting while working with binary outputs. Let $\phi : [k] \to \{0, 1, *\}$ represent a function that maps labels to $\{0, 1, *\}$, which we term a *collapsing map*. We define a family of collapsing maps $\Psi$ as a set of collapsing maps. The definitions of labeled trees will be the only distinction from the regular definition of multiclass Littlestone dimension; and every node in addition to an example will also have a collapsing map assigned to it.

**Definition 4.1** ($\Psi$-labeled binary tree)**.** *A complete $\Psi$-labeled binary tree of depth $b$ with label set $[k]$ and mapping set $\Psi$ on input space $\mathcal{X}$ consists of a complete binary tree of depth $b$ with the following labels:*

1. *Every node of the tree other than the leaves is labeled by an example $x \in \mathcal{X}$, and a collapsing map $\phi \in \Psi$.*

2. *The left and right edges going from any parent node to its two children are labeled by $0$ and $1$ respectively.*

3. *The leaf nodes of the tree are unlabeled.*

Where the input space, label space and mapping set are obvious, we will omit them and simply refer to a complete $\Psi$-labeled binary tree.

**Definition 4.2.** *Consider a root-to-leaf path in a complete $\Psi$-labeled binary tree described as an ordered sequence $S = ((x_0, \phi_0, y_0), \ldots, (x_{b-1}, \phi_{b-1}, y_{b-1}))$, where each $x_i \in \mathcal{X}$ is an input, $\phi_i$ is a collapsing map, and $y_i \in \{0, 1\}$ is an edge label. We say that this path is realized by a function $f \in H$ if $y_i = \phi_i(f(x_i))$ for every triple in the ordered sequence $S$.*

We can now define what it means for a class of functions to $\Psi$-shatter a complete $\Psi$-labeled binary tree.

**Definition 4.3** ($\Psi$-shattering). *We say that a complete $\Psi$-labeled binary tree of depth $b$ with label set $[k]$ is $\Psi$-shattered by a hypothesis class $H$ if for all $2^b$ root-to-leaf sequences $S$ of the tree, there exists a function $f \in H$ that realizes $S$.*

Finally, we are in a position to define the $\Psi$-Littlestone dimension.

**Definition 4.4** ($\Psi$-Littlestone dimension). *The $\Psi$-**Littlestone dimension** $\Psi_{LD}(H)$ of a hypothesis class $H$ is defined to be the maximum depth $b$ such that there is a complete $\Psi$-labeled binary tree of depth $b$ that is $\Psi$-shattered by $H$. If no maximum exists, then we say that the $\Psi$-Littlestone dimension of $H$ is $d = \infty$.*

## 4.2 Properties of $\Psi$-Littlestone dimension

In this section, we begin our investigation of the $\Psi$-Littlestone dimensions by discussing a few simple and useful properties that they have.

We define three important families of collapsing maps $\Psi^N$, $\Psi^{bin}$ and $\Psi^B$ that will play an important role in our results. Consider a collapsing map $\phi_{w,w'}$ defined by $\phi_{w,w'}(\ell) = 0$ if $\ell = w$, $\phi_{w,w'}(\ell) = 1$ if $\ell = w'$, and $\phi_{w,w'}(\ell) = *$ otherwise. Then, $\Psi^N$ is defined to be $\{\phi_{w,w'} | w \neq w', w, w' \in [k]\}$. Similarly, let $\phi_i$ be a collapsing map that maps a label to the $i^{th}$ bit of its $\log(k+1)$ bit binary expansion. Then, $\Psi^{bin}$ is defined to be $\{\phi_i \mid i = 1, \cdots, \log(k+1)\}$. Finally, $\Psi^B$ is defined as the family of all collapsing maps from $[k]$ to $\{0, 1, *\}$.

For any hypothesis class $H$, we show the following properties of $\Psi_{LD}^N(H), \Psi_{LD}^{bin}(H)$ and $\Psi_{LD}^B(H)$.

1. For all hypothesis classes $H$, $MLS(H) = \Psi_{LD}^N(H)$.
2. For all hypothesis classes $H$, the $\Psi^{bin}$-Littlestone dimension upper bounds the maximum Littlestone dimension over its binary restrictions.
3. For all hypothesis classes $H$, $\Psi_{LD}^N(H) \leq \Psi_{LD}^{bin}(H) \leq \Psi_{LD}^B(H)$.

The proofs of these facts can be found in the supplementary material. These facts show the expressive power of the $\Psi$-Littlestone dimension.

# 5 Proof of Theorem 3.3

In this section, we use the concept of $\Psi$-Littlestone dimension to prove Theorem 3.3.

## 5.1 Sauer's lemma for multiclass Littlestone dimension

In this section, we will describe a version of Sauer's Lemma that will suffice for our application. This argument is essentially due to Rakhlin et al. [20]. Theorem 7 in that paper states a Sauer's lemma style upper bound for a quantity they introduce called '0-cover function', for hypothesis classes with bounded 'sequential fat-shattering dimension'. We show that this argument applies almost verbatim for hypothesis classes with bounded multiclass Littlestone dimension.

### 5.1.1 0-Cover function

We start by recalling the definition of 0-cover from Rakhlin et al.

**Definition 5.1** (output-labeled trees, input-labeled trees). *A complete output-labeled binary tree of depth $b$ with label set $[k]$ is a complete binary tree of depth $b$ such that every node of the tree is labeled with an output $\in [k]$. A complete input-labeled binary tree of depth $b$ with input set $\mathcal{X}$ is a complete binary tree of depth $b$ such that every node of the tree is labeled with an input in $\mathcal{X}$.*

The convention we will use is that output and input-labeled binary trees have root at depth $1$ (as opposed to io-labeled trees and $\Psi$-labeled trees, where we use the convention that root has depth $0$). Consider a set $V$ of complete output-labeled binary trees of depth $b$ with label set $[k]$. Consider a hypothesis class $H$ consisting of functions from input space $\mathcal{X}$ to label set $[k]$. Fix a complete input-labeled binary tree $z$ of depth $b$ with input space $\mathcal{X}$ and a complete output-labeled tree $v \in V$.

**Definition 5.2.** *We say that a root-to-leaf path $A$ in $z$ **corresponds** to a root-to-leaf path $B$ in $v$ if for all $1 \leq i \leq b - 1$, if node $i + 1$ in $A$ is the left child of node $i$ in $A$, then node $i + 1$ in $B$ is the left child of node $i$ in $B$ and likewise for the case where node $i + 1$ is the right child of node $i$.*

**Definition 5.3.** *Let $A$ be a root-to-leaf path in $z$ and let the the labels of the nodes in $A$ be $(x_1, \cdots, x_b)$ where $x_i \in \mathcal{X}$. The function $f \in H$ applied to $A$, denoted by $f(A)$, is the sequence $(f(x_1), \cdots, f(x_b))$.*

**Definition 5.4** (0-cover, [20])**.** *We say that $V$ forms a **0-cover** of hypothesis class $H$ on tree $z$ if, for every function $f \in H$ and every root-to-leaf path $A$ in $Z$, there exists a complete output-labeled tree $v \in V$, such that for the corresponding root-to-leaf path $B \in v$ with the labels of nodes in $B$ denoted by a tuple $C \in [k]^b$ (call this the **label sequence** of $B$), we have that $f(A) = C$.*

**Definition 5.5** (0-cover function, [20])**.** *We will use $N(0, H, z)$ to denote the size of the smallest 0-cover of hypothesis class $H$ on tree $z$. Let $T_b^{\mathcal{X}}$ be the set of all complete input-labeled binary trees of depth $b$ with input space $\mathcal{X}$. Then, the **0-cover function** $N(0, H, b)$ of hypothesis class $H$ is defined as $\sup_{z \in T_b^{\mathcal{X}}} N(0, H, z)$.*

### 5.1.2 Statement of theorem

The following theorem is essentially Theorem 7 in Rakhlin et al. (with multiclass Littlestone Dimension instead of Sequential Fat Shattering Dimension).

**Theorem 5.6.** *Let hypothesis class $H$ be a set of functions $f : \mathcal{X} \to [k]$. Let the multiclass Littlestone Dimension of $H$ be $d$. Then, for all natural numbers $n \geq d$,*

$$N(0, H, n) \leq \sum_{i=0}^{d} \binom{n}{i} k^i \leq \left( \frac{ekn}{d} \right)^d$$

*and for all natural numbers $n < d$, $N(0, H, n) \leq (k + 1)^n$.*

The proof of this theorem proceeds via an inductive argument, and can be found in the full version of the paper.

## 5.2 Lower bound for $0$-cover function

To complement the upper bound given by our variant of Sauer's Lemma, we give a lower bound showing that the 0-cover function must grow exponentially in the $\Psi^B$-Littlestone dimension of a class.

**Lemma 5.7.** *Let the $\Psi^B$-Littlestone Dimension of hypothesis class $H$ be $d$. Then,*

$$N(0, H, d) \geq 2^d.$$

The proof can be found in the full version of the paper.

## 5.3 Putting the pieces together

In this section, we prove Theorem 3.3 using the techniques we have built up.

Let $H$ be a hypothesis class with $MLS(H) = d$ and let $H|_1, \cdots, H|_{\log(k+1)}$ be its binary restrictions with Littlestone dimensions $d_1, \cdots, d_{\log(k+1)}$. Let the $\Psi_{LD}^B(H)$ be $d_B$. Then by the discussion in Section 4.2, we have that $d \leq d_B$. Additionally, using Lemma 5.7 and Theorem 5.6 with $n = d_B \geq d$, we have that

$$2^{d_B} \leq N(0, H, d_B) \leq \left( \frac{ekd_B}{d} \right)^d. \tag{1}$$

We will use the fact that for all positive real numbers $x, y$, $\ln x \leq xy - \ln(ey)$. Fix some constant $y < \ln 2$ to be chosen later. We can simplify equation 1 with the following chain of inequalities.

$$2^{d_B} \leq \left(\frac{ekd_B}{d}\right)^d \implies d_B \ln 2 \leq d\left(\ln\left(\frac{d_B}{d}\right) + \ln(ek)\right)$$

$$\implies d_B \ln 2 \leq d\left(y \cdot \frac{d_B}{d} - \ln(ey) + \ln(ek)\right)$$

$$\implies d_B \leq \frac{1}{\ln 2 - y} d \ln\left(\frac{k}{y}\right).$$

Setting $y = \frac{1}{5} < \ln 2$, we get that

$$d_B \leq 6d\ln(k+1).$$

Finally, by the discussion in Section 4.2, we know that the $\Psi_{LD}^{bin}(H)$ is upper bounded by $d_B$ and lower bounded by the maximum Littlestone dimension of the binary restrictions of $H$. Then,

$$\max_{i=1,\cdots,\log(k+1)} d_i \leq \Psi_{LD}^{bin}(H) \leq d_B \leq 6d\ln(k+1).$$

This proves the theorem.

**Funding and acknowledgements:** Mark Bun was partially supported by NSF grants CCF-1947889 and CNS-2046425. Satchit Sivakumar was partially supported by NSF grant CNS-2046425. Marco Gaboardi was supported by the National Science Foundation under awards CNS2040249 and CNS2040215. In addition, this work was partially supported by Cooperative Agreement CB20ADR0160001 with the Census Bureau. The views expressed in this paper are those of the authors and not those of the U.S. Census Bureau or any other sponsor.

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
