# OpenReview forum: "Multiclass versus Binary Differentially Private PAC Learning"
_NeurIPS.cc/2021/Conference — NeurIPS 2021 Poster_

### Official Review · Reviewer_ErcE · 2021-07-13

**Rating:** 7
**Confidence:** 3

**Summary:**

This paper studies the problem of multiclass differentially private PAC learning. It provides a generic reduction from multiclass differentially private PAC learning to binary private PAC learning, that is intuitively based on the decomposition of an integer $k$ into $\log(k)$ bits. As a result, the contribution of the paper is a private multiclass learning algorithm (with sample complexity polynomial in the multiclass Littlestone dimension and poly-logarithmic in the number of labels-classes). This work is closely related to recent progress in the binary classification regime towards the connections between differentially private learnability and online learning.

**Limitations And Societal Impact:**

The authors propose some potential future directions which correspond to some limitations of the existing work and hence address adequately this task. I do not think there is any potential negative societal impact as a byproduct of this work.



**Main Review:**

This work studies multiclass differentially private PAC learning. The main question asked in this work is whether one can obtain multiclass DP learning results based on the existing binary DP learning results.

At a technical level, the paper relies on two known facts: a (simple) reduction from multiclass learning to binary and recent advances on (binary) DP PAC learning. Specifically, it is based on a well-known reduction from a given multiclass learning problem to a collection of binary sub-problems (that intuitively corresponds to the decomposition of an integer $k$ into $\log(k)$ bits) and on a recent result that a binary class is privately learnable using $\tilde{O}(d^6)$ samples, where $d$ is the Littlestone dimension of the class.

Using the aforementioned reduction, the authors obtain that if all the binary sub-classes are privately learnable, then the original multi-label class is privately learnable too. The main technical contribution of this work is the statement that if a multi-valued class has bounded multiclass Littlestone dimension, then all of its binary sub-classes have bounded binary Littlestone dimension (recall that the finiteness of the (binary) Littlestone dimension of a binary class implies that the class is privately learnable). The authors provide a tight quantification of this statement.

The tools behind the proof lie mainly in the well-known work on standard multiclass PAC learning and the $\Psi$-dimension. Specifically, this work provides an extension of this capacity measure to the online setting: they show that one can choose a mapping $\Psi$ so that the $\Psi$-dimension corresponds to the multiclass Littlestone dimension while another choice of mapping can yield the maximum Littlestone dimension of the binary associated classes.

Strengths: The problem studied in this paper is a natural extension of the existing literature. The connection with the previous results is clear and the provided technical result between multiclass and binary Littlestone dimension is nice. The sample complexity result has the additional advantage of being modular, in the sense that future improvements for the binary classification task can be directly applied to the multiclass case. The online analogue of the $\Psi$-dimension is novel and seems natural and useful.

Weaknesses: The main novel technical contribution lies only in the introduction of the online version of the $\Psi$-dimension in order to obtain the result that finite multiclass Littlestone dimension implies control to the Littlestone dimension of the binary sub-classes. The other two results (the reduction and the sample complexity improvement) are based on existing results: the reduction is essentially similar to the non-private setting and the proof has the additional privacy part (which is not technically challenging) and the sample complexity bound is a direct application of the binary $\tilde{O}(d^6)$ result thanks to the reduction.

Previous Work: It is clearly presented how this work differs from the existing results and how existing results are used.

Correctness: As far as I can tell, yes.

Clarity: The paper is well-written and the results are clearly stated.

Significance: The results of the paper are interesting; this work is based on recent advances in (mainly binary and DP) learning theory and provides understanding towards the multiclass classification case.

**Time Spent Reviewing:**

5-6

---

### Official Review · Reviewer_T8vG · 2021-07-15

**Rating:** 7
**Confidence:** 4

**Summary:**

The paper gives a generic reduction from from private binary PAC learning to private multi-class PAC learning. This paper improves the previous known results for multi class PAC learning in terms of both the dependence on the multi class Littlestone dimension (parameter which characterizes private PAC learning) and the number of classes.

**Limitations And Societal Impact:**

Yes.

**Main Review:**

The paper gives a generic reduction from from private binary PAC learning to private multi-class PAC learning and gets exponential improvement compared to previous results in terms of both number of classes and multi class Littlestone dimension. Private learning has been gaining considerable interest and is an important topic. This paper not only gets improved bounds but also since it generically converts guarantees from binary classification to multi-class classification for private learning and would also hold for future private binary classification results. The main technical contribution of the paper is to show that Littlestone dimension of the function class consisting of each bit of the function class is upper bounded by the multi class Littlestone dimension of the class. I think this result is interesting in itself and might be more useful generally. As the paper claims, the techniques in proving this result mostly follows from previous work in the supervised learning setting but, the generic reduction and using this connection is quite interesting and useful.

The paper is generally well written and easy to follow.

**Time Spent Reviewing:**

4

---

### Official Review · Reviewer_8M5o · 2021-07-22

**Rating:** 7
**Confidence:** 3

**Summary:**

This work studies the connection between the differential privacy of binary learning and multiclass learning. The authors first show a simple reduction from multiclass to binary to private learning, meaning that they show that for any $f$ if there is an algorithm that private learns each binary function of $f$ (meaning learns each bit of $f$) then there is a private algorithm for $f$. Furthermore, the authors show tight bounds on Littlestone of multiclass and binary learning. They prove that if a multiclass hypothesis class has Littlestone dimension $d$ then each binary restriction of the same class has Littlestone dimension bounded by $\sim 6d\log k$. The authors provide upper and lower bounds for this claim. Moreover, the authors provide an upper bound to the sample complexity or privately learning a multiclass hypothesis relatively to the Littlestone dimension of this class.

**Limitations And Societal Impact:**

The authors adress the societal impact of their work.

**Main Review:**

First of all, this work is well-organized and well-written and establishes very basic and fundamental results in differential privacy in the multiclass regime. The multiclass setting is a natural generalization from binary learning and providing upper bounds can be very challenging even in the simple PAC learning. Providing bounds on the binary restriction of a class $H$ is sometimes more difficult than proving for the multiclass problem (i.e., for linear classifiers the problems become non-separable in the binary restriction). Therefore, I find Theorem 1.3 very interesting. A weakness of this work is as the authors acknowledge, their strategy is similar to [5]. Overall, I recommend for acceptance.

**Time Spent Reviewing:**

3-4

---

### Author Response · Authors · 2021-08-10
**Response to Reviewers**

We thank the reviewers for their conscientious reading and evaluation of our work.

Since the submission deadline, we were actually able to show that the multiclass Littlestone dimension of a hypothesis class with $k$ labels cannot be too much larger than the maximum Littlestone dimension over its binary restrictions (that is if the latter is $d$, the former is at most $d \log k$; we also showed the tightness of this result). Put together with the tight result that the maximum Littlestone dimension over the binary restrictions of a hypothesis class cannot be more than a logarithmic in $k$ factor larger than its multiclass Littlestone dimension (as already shown in the submission), this completes the account of the relationship between the multiclass Littlestone dimension of a hypothesis class and the maximum Littlestone dimension of its binary restrictions.

---

### Decision · Program_Chairs · 2021-09-27

**Decision:**

Accept (Poster)

**Comment:**

This is an elegant paper establishing fundamental results for multiclass differentially private PAC learning.  The reviewers found the paper well written and the results interesting.